# Bifunctional Hot Water Vapor Template-Mediated Synthesis of Nanostructured Polymeric Carbon Nitride for Efficient Hydrogen Evolution

**DOI:** 10.3390/molecules28124862

**Published:** 2023-06-20

**Authors:** Baihua Long, Hongmei He, Yang Yu, Wenwen Cai, Quan Gu, Jing Yang, Sugang Meng

**Affiliations:** 1College of Material and Chemical Engineering, Pingxiang University, Pingxiang 337055, China; 2Key Laboratory of Applied Surface and Colloid Chemistry, Ministry of Education, School of Chemistry and Chemical Engineering, Shaanxi Normal University, Xi’an 710062, China; 3College of Health Science and Environmental Engineering, Shenzhen Technology University, Shenzhen 518118, China; 4Key Laboratory of Green and Precise Synthetic Chemistry and Applications, Ministry of Education, Huaibei Normal University, Huaibei 235000, China

**Keywords:** carbon nitride, nanostructured, water vapor, H_2_ evolution

## Abstract

Regulating bulk polymeric carbon nitride (PCN) into nanostructured PCN has long been proven effective in enhancing its photocatalytic activity. However, simplifying the synthesis of nanostructured PCN remains a considerable challenge and has drawn widespread attention. This work reported the one-step green and sustainable synthesis of nanostructured PCN in the direct thermal polymerization of the guanidine thiocyanate precursor via the judicious introduction of hot water vapor’s dual function as gas-bubble templates along with a green etching reagent. By optimizing the temperature of the water vapor and polymerization reaction time, the as-prepared nanostructured PCN exhibited a highly boosted visible-light-driven photocatalytic hydrogen evolution activity. The highest H_2_ evolution rate achieved was 4.81mmol∙g^−1^∙h^−1^, which is over four times larger than that of the bulk PCN (1.19 mmol∙g^−1^∙h^−1^) prepared only by thermal polymerization of the guanidine thiocyanate precursor without the assistance of bifunctional hot water vapor. The enhanced photocatalytic activity might be attributed to the enlarged BET specific surface area, increased active site quantity, and highly accelerated photo-excited charge-carrier transfer and separation. Moreover, the sustainability of this environmentally friendly hot water vapor dual-function mediated method was also shown to be versatile in preparing other nanostructured PCN photocatalysts derived from other precursors such as dicyandiamide and melamine. This work is expected to provide a novel pathway for exploring the rational design of nanostructured PCN for highly efficient solar energy conversion.

## 1. Introduction

The hydrogen evolution via photocatalytic water-splitting is potentially an efficient strategy to store clean energy and alleviate emerging energy issues in the future [1,2,3,4]. Polymeric carbon nitride (PCN) has long been proven to exhibit vast potential to achieve this magnificent goal [5,6,7,8]. Unfortunately, this stringent goal is greatly hindered by the low efficiency of bulk PCN due to inherent drawbacks, such as inferior separation and transfer for the photoexcited charge carriers, limited visible light absorption, extremely low specific surface area, and finite active sites [9,10,11,12]. In this regard, plenty of intelligent strategies have been developed to address the shortcomings mentioned above through the introduction of element doping or functional groups [13,14,15,16], modifications with defects or vacancies [17,18,19], regulating the nanostructure [20,21,22,23], adjusting morphology [24,25,26,27], or coupling with other semiconductors for heterojunctions and so forth [28,29,30,31,32]. Among these strategies, the nanostructure embedded in the PCN framework was demonstrated to be a simple and valid method for strikingly promoting the photocatalytic activity of PCN in many aspects [33]. In general, template methods such as hard-templating, soft-templating, and gas-templating were used to create the nanostructures, which resulted in a larger BET specific surface area, more active site quantity and improved the separation efficiency of photo-excited charged carriers. 

Among these, the gas-templating method for nanostructure engineering was widely developed because it can circumvent not only the complexity of operations but also the use of extremely toxic chemical reagents that exist in the hard-templating and soft-templating methods. Additionally, this gas-templating method has the advantage of being simple, cost-effective, template-free, and suitable for large-scale synthesis. Normally, the gas-templating approach is primarily classified into two categories. The first is the self-induced gaseous templating method, which engineers the porous structure in PCN by using the self-generated gas as a template. For example, Tang et al. successfully obtained porous PCN with an extraordinary hydrogen evolution rate by direct thermal polymerization of urea at high temperatures in the air without using any additional chemical reagents. The porous structure is created by the emission of a large amount of ammonia gas and water vapor due to the existence of an oxygen element in the urea precursor, which is supposed to act as the gaseous template [34]. In another typical work, Chen et al. created nanoporous PCN with an increased BET specific surface area and pore volume via one-step polymerization of the single urea with self-supported gas, and the resulting nanostructured photocatalyst demonstrated a much higher hydrogen evolution rate [35]. The authors believe that the water vapor bubble served as a gaseous template in the proposed nanoporous PCN. Meanwhile, our groups also developed some simple sulfur-containing organic and inorganic compounds, such as trithiocyanuric acid, thiourea, guanidine thiocyanate, or ammonium thiocyanate, to serve as the unitary precursor for the one-pot production of nanoporous PCN at a high temperature [36,37,38,39]. The self-generated sulfur-containing gases produced at high temperatures are thought to be responsible for forming the nanostructure in PCN. However, the above synthesis method needed a high polymerization temperature to produce plenty of gas function as a gaseous template and also induced the destruction of the integrated PCN framework to some extent. The other method uses extra chemical reagents as dynamic gas-bubble templates. To date, specific types of chemical reagents, including NH_4_Cl, (NH_4_)_2_CO_3_, NaHCO_3_, (NH_4_)_2_SO_4_, (NH_4_)_2_S_2_O_8_ and sublimed sulfur, were intensely used as dynamic gas-bubble templates to promote the formation of nanostructured PCN [40,41,42,43,44,45,46,47,48,49]. During high-temperature calcining, the above chemical reagents can thermally decompose into a large number of gases as dynamic bubble templates, whose emissions induce the porous nanostructure in the PCN. Regrettably, the gases (NH_3_, HCl, and SO_2_) released by the above thermal decomposition of chemical reagents cannot be reused and are also sometimes detrimental to the environment, even though the method possesses the benefits of being low cost and easy to operate. Therefore, the exploration of green and pollution-free gas-bubble template methods to realize the synthesis of advanced nanostructured PCN is still of great urgency and interest. 

Recently, the use of water vapor (completely green and abundant in the earth) for the pretreatment or preparation of the catalyst has been reported, and the as-prepared catalysts show astonishingly enhanced catalytic activity [17,50,51]. For instance, Huang et al. reported that an increase in the grain-boundary density in the Pd/Al_2_O_3_ catalyst is achieved by simple water vapor pretreatment and oxidation. The pretreatment catalyst showed a twelve-fold increase in methane oxidation compared to conventional pretreatments [50]. Yang et al. prepared the few-layered nanostructured PCN by using the bulk PCN and water vapor as the precursor and gas-bubble templates, respectively. The emissions of CO, H_2_, and NO gas produced from the C/N-steam reforming reactions also played an important role in the formation of nanostructures in the PCN. Thus, the water vapor showed a dual function in their work, one was the dynamic gas-bubble template, and the other was a green initiator reagent for chemical etching [17]. However, their synthesis of nanostructured PCN required high-quality bulk PCN to serve as the precursor, which was relatively complex and time-consuming, although the method was demonstrated to be facile, green, and easy to scale up. 

In this work, we proposed a one-step route for synthesizing nanostructured PCN to efficiently enhance its photocatalytic activity by judiciously introducing hot water vapor into the direct thermal polymerization of the guanidine thiocyanate precursor. The hot water vapor served a dual function as a dynamic gas-bubble template and an assisted chemical etching reagent in this synthesis. In particular, the supply of the water vapor is continuous throughout the whole synthesis procedure. We investigated the effect of the polymerization reaction time and the temperature of the water vapor on the morphology, structure, and optical/photoelectric properties of the as-prepared nanostructured PCN. Benefitting from the synchronous nanostructure and carbon vacancies embedded in the PCN, the optimized water-vapor treatment PCN was four times more effective than that of the bulk PCN treatment for photocatalytic hydrogen evolution. Moreover, this hot water vapor dual-function mediated method was also successfully extended to other PCN precursors (melamine and dicyandiamide) to obtain their corresponding derived nanostructured PCNs. The detailed synthesis processes were described, and comprehensive characterizations were conducted to elucidate the enhanced photocatalytic hydrogen evolution mechanism.

## 2. Results and Discussion

### 2.1. Morphology and Texture

The nanostructured photocatalysts were prepared via direct thermal polymerization and simultaneous chemical etching of the guanidine thiocyanate precursor with the assistance of N_2_ flow, carrying the special temperature of the water vapor. We investigated the variability in water-vapor amounts in detail by accurately controlling the temperature of the water vapor and the reaction times during the polymerization process. Unfortunately, the guanidine thiocyanate precursor was burned off at water-vapor temperatures above 60 °C, and thus no photocatalysts were left in our present experimental conditions. The possible explanation is that the PCN or the intermediates of PCN formed during the polymerization process were completely etched by the excess water vapor. In addition, no photocatalysts were collected at the temperature of 60 °C water vapor for 4 h. The above synthesis results suggested that the selection of water vapor temperatures and reaction times is a key factor in providing an optimal amount of water vapor for preparing and modulating the nanostructured PCN.

The changes in the microstructure of the as-prepared photocatalysts were first characterized by SEM and TEM measurements. SEM observations revealed that CGS-CN displayed a compact, thick, and large aggregate morphology (Appendix A). The GS-CN-25 had a similar morphology to that of the CGS-CN due to the insufficient water vapor and short etching reaction time provided in this preparation system. In contrast, both GS-CN-60 and GS-CN-25-4h photocatalysts obtained upon increasing the temperature of the water vapor or prolonging the etching reaction time exhibited looser, thinner, and smaller aggregates. Excitingly, some nanosheets and pores were observed at the surface of the GS-CN-60 photocatalyst, and its contrast also illustrated its maximal BET specific surface area as evidenced by the BET analyses described below. The possible reason for this evolution is due to the significant contribution from the water vapor, which not only avoids the compact and large aggregates of CGS-CN but also its ability to etch the sheets to generate pores via the large number of gases (H_2_, CO, CO_2,_ and NH_3_) released during the simultaneous polymerization and etching procedure [51]. In other words, the released gases are able to explode many “tiny bombs” on the thick chunks of the CGS-CN, which leads to the generation of relatively loose, thin, small aggregates and even porous structural characteristics on the GS-CN-60.

TEM results further substantiated this visible evolution, displaying the bulk CGS-CN aggregates’ gradual evolution into thin and semitransparent nanosheets along with certain surface pores on the GS-CN-60 photocatalyst by the hot water vapor dual -unction mediated method, as seen in Figure 1a,b. AFM topography and height profiles (Figure 1c–f) demonstrated that the GS-CN-60 existed as nanosheets with thicknesses ranging from 2 to 6 nm, while the average thickness of the bulk CGS-CN was around 20 nm. The unique morphological characteristics of the GS-CN-x photocatalysts directly affected their textural properties, as evidenced by the BET test. 

BET confirmed that the specific surface area gradually increased from CGS-CN, GS-CN-25, and GS-CN-25-4h to GS-CN-60, as shown in Figure 2. The GS-CN-60 had the largest surface area with 34.1 m^2^∙g^−1^, which was about 1.7, 1.9, and 7.1 times higher than those of the GS-CN-25-4h, GS-CN-25, and bulk CGS-CN. Nevertheless, the surface area of GS-CN-25-4h (20.4 m^2^g^−1^) only weakly increased compared to that of the GS-CN-25 (17.9 m^2^g^−1^). These results reflected that the hot water vapor assisted by the dual function mediated strategy exerted the most significant impact on the surface area. Moreover, the adsorption–desorption isothermal curves for GS-CN-x photocatalysts all exhibited typical type IV with an H_3_-type hysteresis loop and an enlarged pore volume (Figure 2), revealing that they possessed representative porous structural characteristics and were in line with the results of the SEM and TEM images. Consequently, the above results implied that the optimization of the nanostructured PCN with plentiful pores, high specific surface area, and expanded pore volume was successfully prepared by this hot water vapor dual-function mediated strategy by simply adjusting the relative amounts of water vapor, which promoted the guanidine thiocyanate precursor for suitable seed nucleation, growth, and synchronous etching during the polymerization processes.

### 2.2. XRD and FTIR Analysis

Figure 3a represents the XRD patterns and diffraction peaks of all the photocatalysts. All tested photocatalysts exhibited two distinct diffraction peaks at around 13.3° and 27.2°, which are indexed to the (100) and (002) planes of hexagonal PCN. These reflections correspond to the in-plane structural heptazine units and interlamellar stacking distance, respectively. The results implied that the GS-CN-x could retain the primary heptazine structure of PCN after the water-vapor treatment reaction. Compared with bulk CGS-CN, the (100) diffraction peak intensity for GS-CN-25 showed no noticeable change, indicating that the in-planar layer size of GS-CN-25 had no significant influence (Appendix A). Nonetheless, the GS-CN-60 and GS-CN-25-4h photocatalysts exhibited a slightly weak diffraction peak at (100), attributed to the decrease in-planar layer size. Similarly, the relative intensity of the (002) diffraction peak for GS-CN-60 and GS-CN-25-4h also became weaker, which is attributed to the selective dismemberment of the PCN framework and the formation of the short-range ordered graphite molecular fragments upon increasing the temperature of the water vapor or prolonging the reaction time [52]. In addition, further observation indicated that the (002) peak position for GS-CN-x showed a progressive upshift compared to that of CGS-CN, highlighting the compacted interlayer stacking distance in the resulting water vapor treatment photocatalysts. The reason for this was that the undulated single layers in CGS-CN were planarized by the water-vapor treatment and thus resulted in a tight stack structure for these GS-CN-x photocatalysts [17]. 

The FT-IR spectra presented in Figure 3b show that the GS-CN-x photocatalysts exhibited the same characteristic vibrational absorption peaks as that of CGS-CN. In any case, the identified peak positions at 810, 1200–1700, and 3000–3500 cm^−1^, respectively, were assigned to the bending vibrations of heptazine units, stretching vibrations of aromatic C-N heterocycles, and stretching vibrations of the -NH_x_ groups. Interestingly, the more vigorous intensity of these peaks in the region of 3100–3400 cm^−1^ for the GS-CN-x photocatalysts compared with that of the CGS-CN is indicative of more adsorbed H_2_O on these GS-CN-x photocatalyst surfaces due to their sizeable open-up surface effects.

### 2.3. XPS and ESR Analysis

XPS further demonstrated the more subtle chemical structure and valence state of the photocatalysts. The XPS test confirmed the presence of carbon, nitrogen, and oxygen elements for all the photocatalysts, as illustrated in Figure 4a. The C 1s and N 1s high-resolution spectra of the GS-CN-x photocatalysts showed the same binding energies as that of the CGS-CN, indicating that the heptazine structure was hardly changed in the rigid water-vapor treatment conditions. The high-resolution C 1s spectrum revealed two labeled peaks with the binding energy centered at 288.0 and 284.6 eV (Figure 4b), which were assigned to sp^2^-hybridized aromatic C atoms (N-C=N) in the heptazine rings and adventitious carbon, respectively. The high-resolution N 1s spectrum could be deconvoluted into four labeled peaks with the binding energy positioned at 398.5, 400.1, 401.2, and 404.0 eV, respectively (Figure 4c). The strongest N 1s peak at 398.5 eV corresponded to sp^2^-hybridized aromatic N atoms (C=N-C) in the heptazine rings. The second strongest N 1s peak at 400.1 eV was indicative of tertiary N atoms from N-(C)_3_ groups. The third N 1s peak at 401.2 eV was attributed to amino functional groups (C-N-H). The weakest N 1s peak at 404.0 eV was caused by π*-excitation. The analysis of the surface C/N atomic ratio results in Appendix A show that these GS-CN-x photocatalysts were deficient in carbon compared to CGS-CN. This result reflected the fact that the carbon vacancies were successfully incorporated into the GS-CN-x framework due to the preferential elimination of carbon atoms in the water-vapor treatment reaction. To distinguish the locations of carbon vacancies in the GS-CN-x framework, the summaries of C and N atomic contents were analyzed and quantified based on the peak area ratio (Appendix A). The atomic percentages of N-(C)_3_ and C-N-H decreased, and that of C=N-C increased in N 1s XPS spectra for these GS-CN-x photocatalysts as compared with those of the bulk CGS-CN (Appendix A), indicating that the elimination of the carbon atoms mainly occurred at the N-(C)_3_ and C-N-H sites to generate carbon vacancies for these GS-CN-x photocatalysts. In addition, the increased atomic percentages of N-C=N in C 1s XPS spectra for GS-CN-x photocatalysts further cross-validate the above-mentioned deduction (Appendix A). Nonetheless, the decreased atomic percentages of C-C/C=C for GS-CN-x photocatalysts could result from the part of graphitic carbon being removed from the surface by the reductive gas of H_2_, which was generated through the water vapor etching reaction [51,53]. 

The direct evidence to confirm the formation of carbon vacancies in GS-CN-x can be further interpreted by EPR. As shown in Figure 4d, the Lorentzian line of bulk CGS-CN showed a feeble signal at g = 2.004. The EPR signal of PCN is stemmed from the unpaired electrons on sp^2^-C atoms of aromatic C-N heterocycles, which leads to structural defects in the PCN framework. Moreover, the signal intensity increases gradually from CGS-CN, GS-CN-25, and GS-CN-25-4h to GS-CN-60. The increase of unpaired electrons on the sp^2^-C atoms for GS-CN-x photocatalysts remarkably strengthened the intensity of the Lorentzian line, firmly showing that the concentration of carbon vacancies in the as-prepared GS-CN-x was increased and controllably tuned [54]. However, the bulk C/N atomic ratio in the EA analysis results (Appendix A) suggested that the bulk C/N atomic ratio for these GS-CN-x photocatalysts was slightly decreased compared to the bulk CGS-CN, signifying that the carbon vacancies were more likely to be near the surface of the water-vapor-etched photocatalysts. 

### 2.4. UV-Visible and PL Analysis

The UV-vis absorption spectra of CGS-CN and GS-CN-x photocatalysts are shown in Figure 5a. The intrinsic absorption edge of the GS-CN-x showed a progressive blue shift compared to that of the bulk CGS-CN, which rendered the enlargement of the intrinsic bandgaps. It is noticeable that GS-CN-60 and GS-CN-25-4h photocatalysts show virtually identical absorption edges and intensities. Furthermore, no evident Urbach tail absorption in the visible-light region for the GS-CN-x was found, indicative of the absence of shallow trap states embedded in the bandgap of GS-CN-x generated by carbon vacancies [18,55]. This also indirectly demonstrated that the following enhancement of GS-CN-x photocatalytic activity was not directly correlated with their optical absorption. The corresponding bandgap energies of the photocatalysts were calculated based on the plots of [F(R)hν]^1/2^ versus hν (Figure 5b) and the bandgap energies of CGS-CN (2.56 eV), GS-CN-25 (2.68 eV), GS-CN-60 (2.73 eV), and GS-CN-25-4h (2.73 eV) were estimated. The broadened bandgap for the GS-CN-x catalysts was firmly demonstrated by a similar tendency in the gradual blue shift in the PL emission spectrum (Figure 5c). This hypochromic-shift phenomenon can be well explained as a consequence of the quantum size effect of the nanostructured materials. The values of the valence band were directly determined by XPS valence band spectroscopy (Figure 5d). The band edge of GS-CN-60 (1.97 eV) was revealed as a 0.21 eV negative shift compared to that of CGS-CN (2.18 eV). The conduction band values of CGS-CN and GS-CN-60 could be calculated as −0.38 eV and −0.76 eV, respectively, according to the Equation: E_CB_ = E_VB_ − E_g_. The corresponding band alignments of CGS-CN and GS-CN-60 are schematically depicted in Appendix A, where the GS-CN-60 exhibited a more thermodynamically enhanced reduction power than that of the CGS-CN, indicative of the more powerful reduction ability of photoexcited electrons at GS-CN-60, which enabled the fast proton reduction in the following photocatalytic hydrogen evolution reaction.

### 2.5. Time-Resolved PL and Photoelectrochemical Analysis

To better understand the recombination kinetics of photoexcited charge carriers, the time-resolved PL decay spectra of CGS-CN and GS-CN-x photocatalysts were recorded (Figure 6a). The fitted PL lifetime-decay curves, according to the two-exponential decay model, revealed that the average radiative lifetimes of CGS-CN, GS-CN-25, GS-CN-60, and GS-CN-25-4h were 6.68, 6.31, 5.42, and 5.47 ns, respectively. All the fitting decay parameters and the pertinent details are summarized in Appendix A. The shortest lifetime of the singlet exciton in GS-CN-60 clearly implied that its depopulation of the excited states primarily occurred through non-radiative pathways, presumably through charge transfer of the electrons to some favorable carbon defect sites, and then promoted the rapid transfer and separation of charge carriers [56,57,58,59]. Concurrently, the change regularity of transient photocurrent responses of the photocatalysts can support the above explanation (Figure 6b). GS-CN-60 gave a higher photocurrent response than those of the other photocatalysts, indicative of its remarkably high charge-carrier separation efficiency. To further understand the dynamic behaviors of photo-generated charge carriers, electrochemical impedance spectroscopy (EIS) was conducted to investigate the properties of the electrode/electrolyte interface, and the result is illustrated in Figure 6c. The GS-CN-60 photocatalyst showed the smallest interfacial charge-transfer resistance due to the synergetic effect of the favorable porous and electronic structures, well in accordance with the photocurrent response. Owning a higher overall electronic conductivity, the photoexcited electron transfer kinetics from the bulk to the interface of GS-CN-60 was faster than that of the other photocatalysts, and therefore it is expected to guarantee high photocatalytic activity. 

### 2.6. Photocatalytic Activities

The photocatalytic activities of the as-prepared photocatalysts were examined by visible-light-induced photocatalytic H_2_ evolution in coexistence with Pt catalyst and triethanolamine (TEOA) sacrificial electron donor. Initially, we investigated the influence of the concentration of TEOA on the rate of H_2_ evolution by GS-CN-60 photocatalyst under visible light irradiation. As shown in Appendix A, the H_2_ evolution rate reached a maximum at a concentration of 10 vol.% TEOA. Hereafter, we chose 10 vol.% TEOA as the sacrificial electron donor for the following experiment. The photocatalytic hydrogen evolution amounts versus irradiation time over the as-prepared photocatalysts were plotted in Figure 7a. The CGS-CN photocatalyst showed the lowest activity, with an H_2_ evolution rate of 1.19 mmol∙h^−1^∙g^−1^. As expected, the photocatalytic H_2_ generation activity was significantly enhanced for the GS-CN-x photocatalysts in comparison with that of CGS-CN, suggesting the positive contribution of the water vapor mediated strategy to the photocatalytic activity. The optimized H_2_ evolution rate of 4.81 mmol∙h^−1^∙g^−1^ was achieved for the GS-CN-60 photocatalyst, which was about four times higher than that of the bulk CGS-CN photocatalyst. This result clearly demonstrated the advantage of the hot water vapor treatment to create nanostructures in PCN. At the same time, the high activity was reproducible for the GS-CN-60 photocatalyst, as demonstrated by its excellent long-term stability over a period of 24 h. The generated amount of H_2_ was about 13.2 mmol∙g^−1^ in the first run and could retain the almost equivalent amount of H_2_ in the subsequent five cycle runs, again revealing the robust stability of the GS-CN-60 for sustainable applications. Above all, the GS-CN-60 maintained a well-retained chemical structure even after five-cycle photocatalytic reactions, as demonstrated in the XRD and FT-IR spectroscopy results, which showed that there was no difference between the used photocatalyst and fresh photocatalyst (Appendix A). The wavelength-dependent apparent quantum yield curve of GSCN-60 matched well with its UV-Vis absorption spectrum, reflecting the light-induced nature of the reaction. Based on the above discussion, we are now in a position to try to understand the probable mechanisms behind the enhanced photocatalytic H_2_ evolution activity of GSCN-60. In all, the exceptionally improved photocatalytic activity of GS-CN-60 was due to the synergistic action of high BET specific surface area in contrast to bulk CGS-CN, an enlarged bandgap, outstanding electron reduction ability, and an elevation of the mobility of photo-excited charge carriers. These results, taken together, definitely favored our proceeding with an investigation of the photocatalytic hydrogen evolution reaction. 

### 2.7. Photocatalytic Activities of the Other Prepared Nanostructured PCN

Last but not least, the generality of the effect of hot water vapor with a dual-function mediated strategy was not exclusive to the guanidine thiocyanate precursor. We also verified the effect of the hot water vapor treatment on the other precursors, such as dicyandiamide and melamine. The SEM and TEM results indicated that these water-vapor-treated CDCDA-CN-x and MA-CN-x photocatalysts also showed loose, thin, small aggregates compared with those of the corresponding bulk PCN (Appendix A), which was well reflected by the gradually increased BET specific surface area results (Appendix A). The XRD, FTIR, UV-Visible absorption, and PL spectra of their bulk PCN and corresponding vapor treatment photocatalysts are shown for comparison (Appendix A), indicating that DCDA-CN-x and MA-CN-x showed a similar variation trend with that of the GS-CN-x. These time-resolved PL, photocurrent-response, and EIS-Nyquist characterization results (Figure 8) revealed that these water-vapor-treatment photocatalysts exhibited greatly increased charge separation and electronic conductibility by virtue of their unique porous and electron structures. As expected, the DCDA-CN-x and MA-CN-x displayed obviously enhanced photocatalytic activity.

The hydrogen-evolution rates of the DCDA-CN-60 and MA-CN-60 were 3.8 and 2.7 times higher than those of the bulk CDCDA-CN and CMA-CN, respectively (Figure 9). The difference in exfoliation behavior of PCN from the corresponding different precursors could well account for the different enhanced factors in the photocatalytic activity of H_2_ evolution.

## 3. Materials and Methods

### 3.1. Materials

The guanidine thiocyanate, melamine, dicyandiamide, triethanolamine (TEOA), and H_2_PtCl_6_∙6H_2_O were of analytical grade and used as received without any further purification. Deionized water was used in all the experiments.

### 3.2. Preparation

2 g Guanidine thiocyanate was thoroughly ground into tiny powders with an agate mortar. After that, the powders were carefully transferred into a porcelain boat and subsequently heated to 550 °C at 5 °C∙min^−1^ with nitrogen flow carrying a specific temperature of water vapor for 2 h in a tubular furnace. To avoid the hot water cooling during the transfer line between the gas-washing bottle and the entrance of the furnace, we twined the heat belt to keep the temperature of the transfer line at 100 °C. During the polymerization process, the hot water vapor, maintained at definite temperatures (25 °C, 60 °C, 80 °C, and 100 °C) via a hotplate magnetic stirrer, was carried into the tubular furnace at the assistance of nitrogen gas with a flow rate of 50 mL/min. Finally, the resulting photocatalysts labeled GS-CN-x (where x represents the temperature of water vapor) were obtained for further use. At the same time, the effect of increasing the water vapor etching reaction time to 4 h for preparing the nanostructure PCN was also explored. The final photocatalyst was denoted as GS-CN-x-y (where x represents the temperature of water vapor, and y represents the reaction time). The schematic diagram of this hot-water-vapor-assisted etching method for nanostructured PCN is illustrated in Appendix A. In comparison, the preparation of bulk PCN as the control photocatalyst was the same as that of GS-CN-x, except for the absence of hot water vapor, which is denoted as CGS-CN for simplicity.

Similarly, we also used other precursors (melamine and dicyandiamide) to witness the same hot-water-vapor-treatment procedure to prepare their corresponding nanostructured PCN. The resulting photocatalysts were denoted as MA-CN-x (melamine as a precursor) and DCDA-CN-x (dicyandiamide as a precursor), where x still represents the temperature of the water vapor. In the meantime, their corresponding bulk PCN photocatalysts as control photocatalysts were also synthesized via calcining pure melamine and dicyandiamide in the absence of hot water vapor, which were denoted as CMA-CN and CDCDA-CN, respectively.

### 3.3. Characterization

The scanning emission microscope measurements were conducted using an FEI Nova Nano SEM 230 (Thermo Fisher Scientific, Waltham, MA, USA) Field Emission Scanning Electron Microscope. Transmission electron microscopy (TEM) images were obtained using an FEI Talos (Thermo Fisher Scientific, Waltham, MA, USA) field emission transmission electron microscope. X-ray photoelectron spectroscopy (XPS) data were collected on a Thermo ESCALAB 250 instrument (Thermo Fisher Scientific, Waltham, MA, USA) with a monochromatized Al Kα line source (200 W). The Fourier transform infrared (FT-IR) spectra were obtained on a Nicolet Nexus 670 FT-IR spectrometer (Thermo Nicolet Co., Madison, USA) in a range from 4000 to 400 cm^−1^, and the photocatalysts were mixed with KBr at a concentration of ca. 1 wt%. Nitrogen adsorption–desorption experiments were performed at 77 K using Micromeritics Tristar II 3020 equipment (Micromeritics, Norcross, GA, USA). The specific surface area was calculated by the Brunauer-Emmet-Teller (BET) method. Elemental analysis (EA) was carried out on an elemental Analyzer (Elementar vario EL cube, Hanau, Germany). X-ray diffraction (XRD) measurements were performed on a Bruker D8 Advance diffractometer (Bruker, Billerica, MA, USA) with Cu Kα1 radiation (*λ* = 1.5406 Å). UV-Vis diffuse reflectance spectra (UV-Vis DRS) were collected on Lambda 650s Scan UV-Visible system (Perkin-Elmer, Waltham, MA, USA) using double beam optic, and Teflon was used as the reflectance standard. Electron paramagnetic resonance (EPR) spectra were tested by Bruker model A 300 spectrometer (Bruker, Billerica, MA, USA). The photoluminescence (PL) spectra were done at room temperature on a Hitachi F-7100 type of spectrophotometer (Hitachi Co., Tokyo, Japan). The time-resolved PL decay spectra were recorded at room temperature on an Edinburgh FI/FSTCSPC FLS-1000 spectrophotometer (Edinburgh, Livingston, UK). The electrochemical measurements were done on a CHI 760E electrochemical workstation (Chenhua Co., Ltd., Shanghai, China) in an electrolytic cell with standard three electrodes. The Ag/AgCl (3M KCl) was used as a reference electrode, and a Pt foil was used as a counter electrode. For the working electrode, the photocatalyst dispersion was dipped into the F-doped tin oxide (FTO) glass with a fixed area of 0.25 cm^2^ and then dried at 120 °C for 2 h to improve adhesion for further use. 

### 3.4. Photocatalytic H_2_ Evolution Experiments

The photocatalytic reactions were performed in an 80 mL volume of Schlenk flask at 1 bar atmospheric pressure of Argon. Typically, 10 mg photocatalyst powder as a photosensitizer was ultrasonically dispersed in 10 vol.% TEOA aqueous solution (10 mL), which was used as the sacrificial electron donor. A 3 wt% Pt as the catalyst was loaded onto the surface of the photocatalyst by the photodeposition approach using H_2_PtCl_6_∙6H_2_O. The reaction system was evacuated and then backfilled with the high-purity Argon gas (99.999%). This process was repeated three times to remove air completely, and at the last cycle, the Schlenk flask was backfilled with 1 bar of the high-purity Argon gas before irradiation under a 300 W Xe-lamp with UV cut-off filter (*λ* > 420 nm). The temperature of the reaction solution was kept at 25 °C by a flow of cooling water. After irradiation, 0.5 mL of the generated gas was extracted per hour and detected by gas chromatography (Fuli, GC-9790Plus, Wenling, China) equipped with a thermal conductive detector (TCD) using Argon as carrier gas. After four hours of terminating the reaction, the reaction system was repeated to evacuate and backfill the Argon gas for the next cycles of the hydrogen-evolution experiments to verify the stability of the photocatalyst. 

The apparent quantum yield (*AQY*) for H_2_ evolution was measured under a monochromatic light with a bandpass filter of 365, 405, 420, and 450 nm, respectively. The intensity of the light was 140, 127, 107, and 144 mW∙cm^−2^ for the 365, 405, 420, and 450 nm monochromatic filters, respectively. The irradiation area was measured as 4.6 cm^2^. According to the amount of hydrogen produced every hour in the photocatalytic reaction, the *AQY* was calculated by the Formula (1):(1)$$\mathrm{AQY}=\frac{{2\;\times \;M\;\times \;N}_{A}}{S\;\times \;P\;\times \;t\;\times \;\lambda /(h\;\times \;c)}$$
where *M* is the mole number of evolved H_2_ (mol), *N_A_* is Avogadro’s constant (6.022 × 10^23^ mol^−1^), *S* is the irradiated area (cm^2^), *P* is the powder density of irradiation light (W∙cm^−2^), *t* is the irradiation time (s), *λ* is the wavelength of the monochromatic light (nm), *h* is the Planck constant (6.626 × 10^−34^ J∙s), *c* is the velocity of light (3 × 10^8^ m∙s^−1^).

## 4. Conclusions

In summary, nanostructured PCN can be successfully prepared by a green and sustainable water-vapor mediated method through one-pot simultaneous polymerization and chemical etching of the PCN precursors directly with hot water vapor. The nanostructured morphology with carbon vacancies can be created and is controllable by controlling the temperature of the water vapor and the reaction time in the synthesis process. Benefitting from the more exposed surface, increased photo-excited electrons reduction ability, and enhanced photo-excited charge carrier transfer and separation efficiency, the GSCN-60 realized substantially improved photocatalytic hydrogen evolution performance than that of the bulk CGS-CN. The present hot water vapor with a dual-function mediated approach could provide a novel pathway for the preparation of nanostructured PCN materials with high photocatalytic performance.

## Figures and Tables

**Figure 1 molecules-28-04862-f001:**
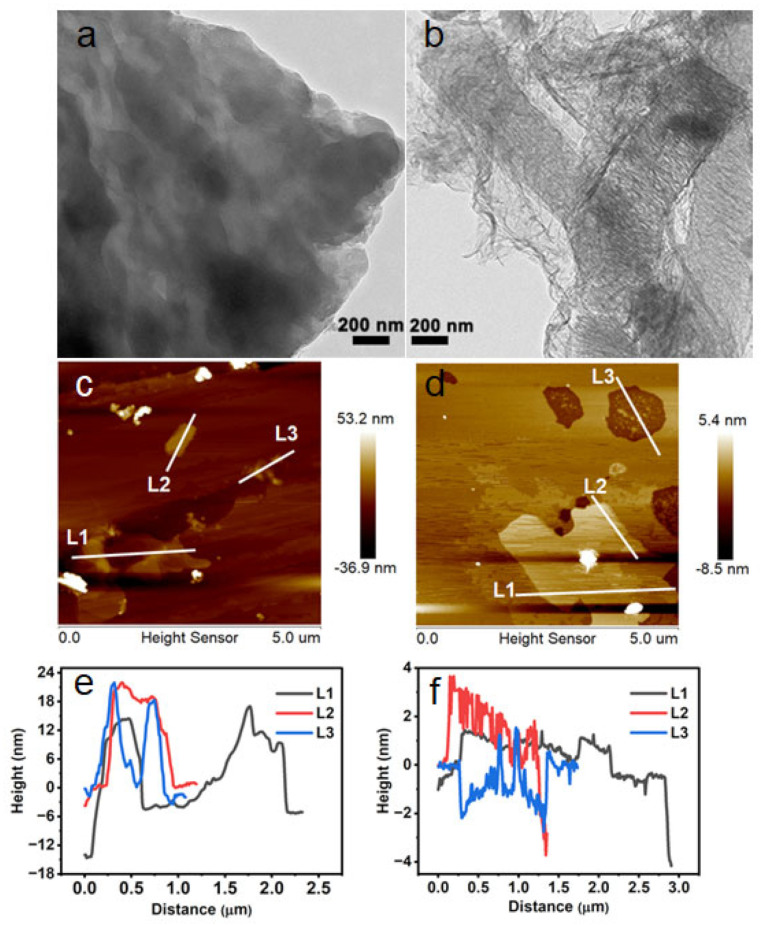
TEM images of: (**a**) bulk CGS-CN and (**b**) GS-CN-60. AFM images and corresponding height profile of: (**c**,**e**) bulk CGS-CN; and (**d**,**f**) GS-CN-60 ((**c**,**d**) corresponds to (**e**,**f**), respectively).

**Figure 2 molecules-28-04862-f002:**
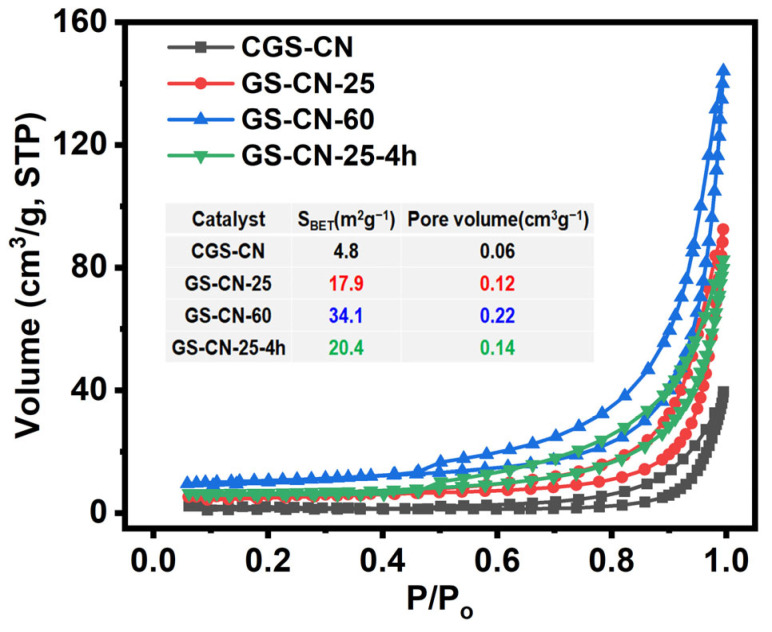
N_2_ adsorption-desorption isotherms of bulk CGS-CN and GS-CN-x photocatalysts.

**Figure 3 molecules-28-04862-f003:**
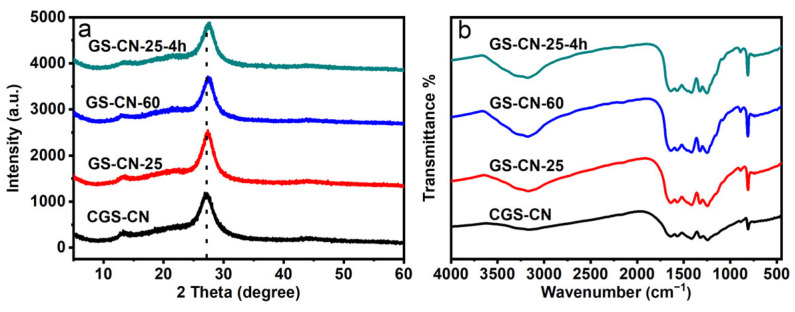
(**a**) XRD patterns of bulk CGS-CN and GS-CN-x photocatalysts; and (**b**) FT-IR spectra of bulk CGS-CN and GS-CN-x photocatalysts.

**Figure 4 molecules-28-04862-f004:**
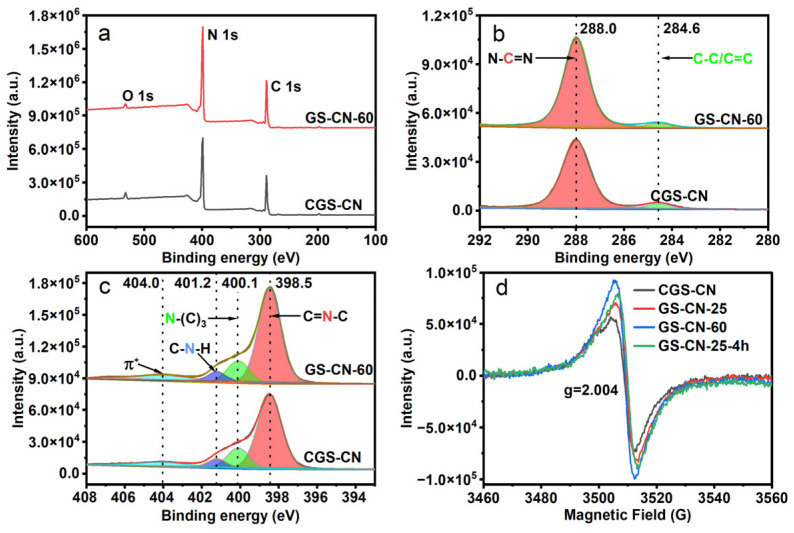
XPS patterns of bulk CGS-CN and GS-CN-x photocatalysts: (**a**) survey patterns; (**b**) high-resolution patterns of C1s; (**c**) high-resolution patterns of N1s; and (**d**) room-temperature EPR spectra of bulk CGS-CN and GS-CN-x photocatalysts.

**Figure 5 molecules-28-04862-f005:**
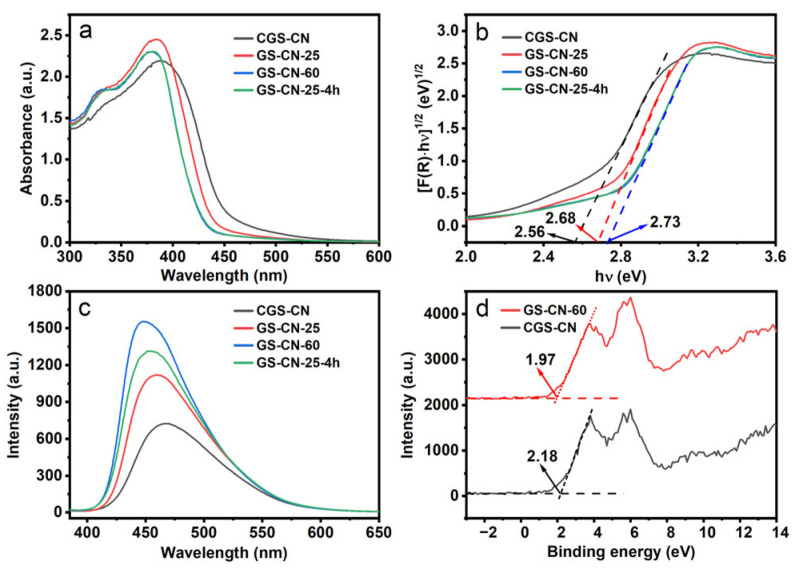
(**a**) UV-Vis diffuse reflectance spectra of bulk CGS-CN, GS-CN-x photocatalysts (noted: the nearly same absorption edge and intensity occurred in GS-CN-60 and GS-CN-25-4h); (**b**) The corresponding Kubelka–Munk transformed spectra of bulk CGS-CN and GS-CN-x photocatalysts; (**c**) FL emission spectra of bulk CGS-CN and GS-CN-x photocatalysts; and (**d**) XPS valence-band spectra of bulk CGS-CN and GS-CN-60 photocatalysts.

**Figure 6 molecules-28-04862-f006:**
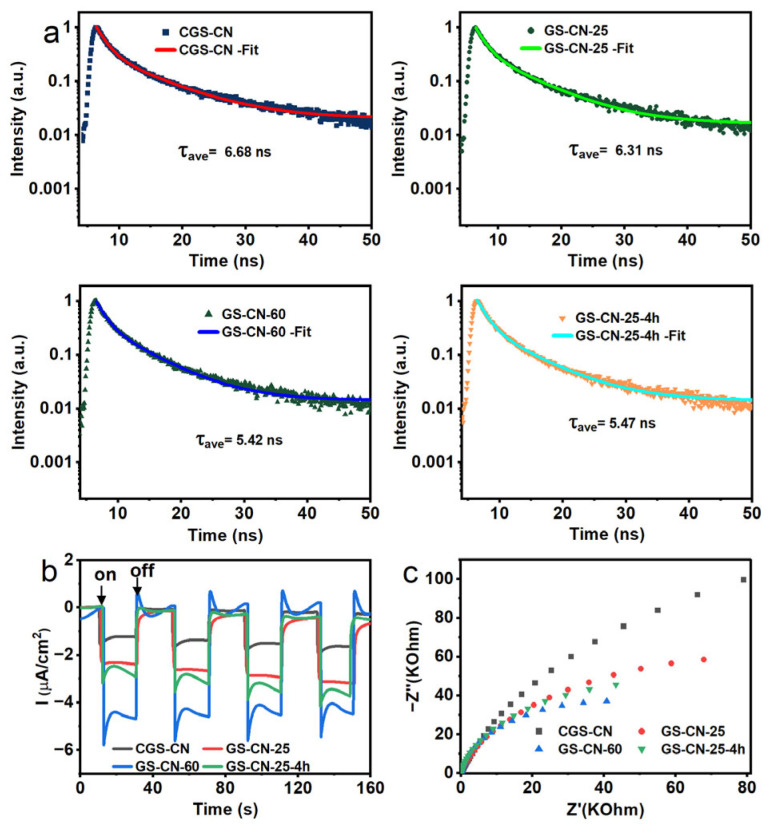
(**a**) Time-resolved PL decay spectra of bulk CGS-CN and GSCN-x photocatalysts kinetics monitored at their maximum emission wavelength (CGS-CN: 470 nm; GS-CN-25: 459 nm; GS-CN-60: 448 nm; GS-CN-25-4h: 453 nm) under 365 nm excitation; (**b**) transient photocurrent responses; and (**c**) EIS Nyquist plots of bulk CGS-CN and GS-CN-x photocatalysts.

**Figure 7 molecules-28-04862-f007:**
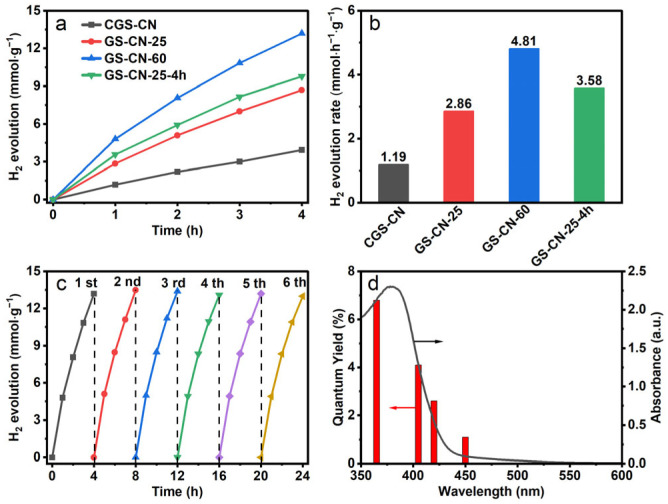
(**a**) Time-dependent evolution of H_2_ produced on bulk CGS-CN and GS-CN-x photocatalysts; (**b**) H_2_ evolution rate in the first hour on bulk CGS-CN and GS-CN-x photocatalysts; (**c**) recycling test of the GS-CN-60 photocatalyst; and (**d**) M of GS-CN-60 under different wavelengths of monochromatic light.

**Figure 8 molecules-28-04862-f008:**
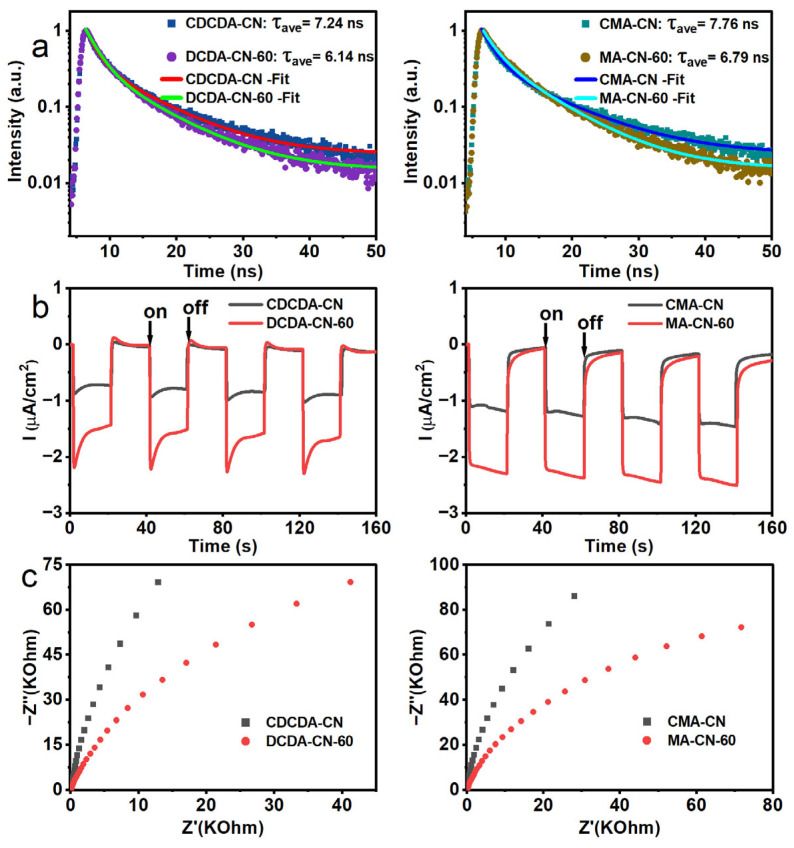
(**a**) Time-resolved PL decay spectra of CDCDA-CN, DCDA-CN-60, CMA-CN and MA-CN-60 photocatalysts kinetics monitored at their maximum emission wavelength (CDCDA-CN: 473 nm; DCDA-CN-60: 454 nm; CMA-CN: 473 nm; MA-CN-60: 445 nm) under 365 nm excitation; (**b**) Transient photocurrent responses; and (**c**) EIS Nyquist plots of CDCDA-CN, DCDA-CN-60, CMA-CN and MA-CN-60 photocatalysts.

**Figure 9 molecules-28-04862-f009:**
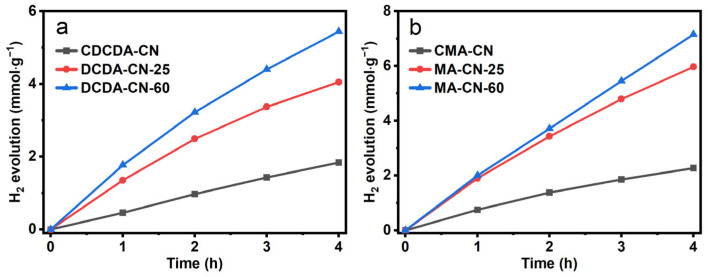
Photocatalytic H_2_ evolution rates of: (**a**) CDCDA-CN and DCDA-CN-x; and (**b**) CMA-CN and MA-CN-x photocatalysts.

## Data Availability

Not applicable.

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
