# Peer review of "Bifunctional Hot Water Vapor Template-Mediated Synthesis of Nanostructured Polymeric Carbon Nitride for Efficient Hydrogen Evolution"

_molecules, 2023, doi:10.3390/molecules28124862_

Round 1

Reviewer 1 Report

Review Report

The work "Bifunctional hot water steam template-mediated synthesis of nanostructured polymeric carbon nitride for efficient hydrogen evolution" by Baihua Long et al. report the one-step green and sustainable synthesis of nanostructured polymeric carbon nitride in the direct thermal polymerization of the guanidine thiocyanate precursor via judiciously introducing hot water steam’s dual function as gas bubble templates along with a green etching reagent. Authors observed that by optimization of the temperature of the water steam and polymerization reaction time, the as-prepared nanostructured polymeric carbon nitride exhibited a highly boosted visible-light-driven photocatalytic hydrogen evolution activity.

The work provide a novel passageway to explore the rational design of nanostructured polymeric carbon nitride for highly efficient solar energy conversion.

The paper reads well and the study is clear and complete. The work is timely for the Molecules community. I recommend to publish it in a present form.

Reviewer 2 Report

In this manuscript the authors report the use of “hot water steam template-mediated synthesis  of nanostructured polymeric carbon nitride” in order to increase the photocatalytic activity of their materials. The protocols of synthesis, characterization of materials and the studies of photocatalytic activity are common and described in the literature. Thus, the novelty of this work in my opinion is questionable, but at the same time I don’t have serious reasons to reject it. I would recommend it for publication after minor revision.

In general, they have studied the photocatalytic dehydration reaction. In this system the polymeric carbon nitride is a photosensitizer and the unit for primary charge separation. Pt is the catalyst for dehydration. This should be clearly stated in the manuscript.  

 The word “steam” means the water vapor and the vapor with small droplets of liquid water. This should be clarified.

Y-axes in several Figures are labeled, as e.g. “Intensity (a.u),” without giving the values. On the Figure 5a is the UV-vis spectra, but neither “concentrations” nor optical pathway are provided.

The possible effect of TEOA addition, the control, on lifetimes is missing (data on Figure 4a).

I don’t understand how 10 mL of solution in 80 mL flask can be irradiated by a large light beam with a surface area 4.6 cm^2. Is it assumed that all light is absorbed?

I don’t understand this conclusion “the more powerful photo-reduction ability of photo-excited electrons, and higher accelerated photo-excited charge carrier transfer and separation substantially improved photocatalytic hydrogen evolution performance.”

This study is dealing with not molecular materials.  It would be more logical to submit this work to the MDPI sister journal “Materials”

Reviewer 3 Report

This manuscript presents an interesting study on the preparation of nanostructured PCN by employing hot water steam templating method during the polymerization of PCN precursors. The study discusses the formation of nanostructured morphology by controlling the temperature of the steam and the reaction time during the polymerisation. The results presented here shows successful fabrication of nanostructured PCN with enhanced photocatalytic performance. This manuscript is well written, and the results presented are supported by experimental evidence and data. Therefore, the manuscript could be considered for publication in ‘molecules’. Please see couple of minor comments below.

1.       Have you attempted different temperature and reaction times to optimize the conditions for the best results? For example, did you try 60oC and 1h or 2h reaction time instead of 4h?

2.       The atomic percentage (of C) estimation from XPS results shows only very minor changes between different samples (40.89 and 40.62%?). Are the numbers significant enough to say the possible carbon vacancies contributes to the nanostructured morphology and enhanced catalytic performance?

3.       Page 2, line 79: “The other one used other extra chemical reagents as dynamic gas bubble 79 templates”. This sentence is not clear. Please rewrite.

4. Page 3, lines 130-132: “The possible explanation…..”. This sentence is also not clear, consider rewriting.

English language is fine, minor editing required
